# Exome-wide analysis implicates rare protein-altering variants in human handedness

Dick Schijven [1,2], Sourena Soheili-Nezhad [1], Simon E. Fisher [1,2] &
Clyde Francks [1,2,3] ✉

Handedness is a manifestation of brain hemispheric specialization. Left-handedness occurs at increased rates in neurodevelopmental disorders. Genome-wide association studies have identified common genetic effects on handedness or brain asymmetry, which mostly involve variants outside protein-coding regions and may affect gene expression. Implicated genes include several that encode tubulins (microtubule components) or microtubule-associated proteins. Here we examine whether left-handedness is also influenced by rare coding variants (frequencies ≤ 1%), using exome data from 38,043 left-handed and 313,271 right-handed individuals from the UK Biobank. The beta-tubulin gene *TUBB4B* shows exome-wide significant association, with a rate of rare coding variants 2.7 times higher in left-handers than right-handers. The *TUBB4B* variants are mostly heterozygous missense changes, but include two frameshifts found only in left-handers. Other *TUBB4B* variants have been linked to sensorineural and/or ciliopathic disorders, but not the variants found here. Among genes previously implicated in autism or schizophrenia by exome screening, *DSCAM* and *FOXP1* show evidence for rare coding variant association with left-handedness. The exome-wide heritability of left-handedness due to rare coding variants was 0.91%. This study reveals a role for rare, protein-altering variants in left-handedness, providing further evidence for the involvement of microtubules and disorder-relevant genes.

Roughly 90% of the human population is right-handed and 10% left-handed[1,2]. Despite some regional and temporal variation, this overall bias is broadly consistent across continents, and has been stable through human history[2–6]. Handedness is a manifestation of brain asymmetry, as right handedness reflects left-hemisphere dominance for control of the preferred hand, and vice versa[7].

Population-level asymmetries of anatomy and function arise in the human brain during fetal development[8–13], and right-lateralized predominance of arm movements has been reported already at ten weeks of gestational age[14]. The early appearance of these asymmetries indicates a genetically regulated program of left-right axis development in the central nervous system[1,15–18]. Consistent with this, left-handedness

has shown heritability of roughly 25% in twin-based analysis[19], and 1–6% in population studies that have assessed the specific contribution of common genetic variants[20,21]. Twin- and family-based studies have also reported heritabilities of up to roughly 30% for measures of structural or functional brain asymmetry, particularly for regions or networks important for language[22–25], which is lateralized to the left hemisphere in most people.

Genome-wide association studies of human handedness in sample sizes of less than 10,000 individuals did not find significantly associated genetic loci[26,27], but two larger-scale studies[20,28] have been performed based on the UK Biobank adult population dataset[29], which included over 30,000 left-handed and 300,000 right-handed

[1]Language & Genetics Department, Max Planck Institute for Psycholinguistics, Nijmegen, The Netherlands. [2]Donders Institute for Brain, Cognition and Behaviour, Radboud University, Nijmegen, The Netherlands. [3]Department of Cognitive Neuroscience, Radboud University Medical Center, Nijmegen, The Netherlands. ✉e-mail: clyde.francks@mpi.nl

individuals. In these larger studies, three or four genomic loci showed statistically significant associations with left-handedness, depending on study-specific inclusion criteria and methods. The implicated genes included *TUBB* which encodes a beta-tubulin component of microtubules, and *MAP2* and *MAPT* which encode microtubule-associated proteins. Microtubules are prominent parts of the cytoskeleton – the framework of protein filaments internal to cells – that contributes to a wide range of processes including cellular growth, division, migration, shape and axis formation, axon outgrowth and intracellular transport[30]. It is not known how microtubules affect inter-individual variation in human handedness, but it has been suggested[17,31] that they may contribute to cellular chirality early in brain development, and thereby to organ-intrinsic formation of the brain's left-right axis (see Discussion).

An even larger genome-wide association meta-analysis study of human handedness has also been performed, including the UK Biobank in addition to many other datasets, for a total of 194,198 left-handed and 1,534,836 right-handed individuals[21]. The greater statistical power of this study resulted in 41 genomic loci being significantly associated with left-handedness, including at least eight that implicated tubulins or microtubule-associated proteins, and other genes involved in axon development and neurogenesis[21].

In addition, genome-wide association scanning using brain imaging data from over 32,000 UK Biobank individuals found 27 independent genetic variants that were significantly associated with different aspects of structural brain asymmetry[17]. Remarkably, almost half of these loci implicated genes that code for tubulins or microtubule-associated proteins[17]. A further study[31] then mapped cerebral cortical structural asymmetry with respect to handedness in 3,062 left-handers and 28,802 right-handers from the UK Biobank, and found that 18 of the 41 handedness-associated genomic loci[21] were also associated with at least one regional cortical asymmetry that is linked to left-handedness. The implicated genes again included several that encode tubulins: TUBB, TUBA1A/TUBA1B/TUBA1C (the latter three genes are clustered together in the genome), TUBB3 and TUBB4A, as well as microtubule-associated proteins MAP2, MAPT and NME7.

All of the large-scale studies mentioned above were based only on common genetic variation, i.e. with allele frequencies in the population of at least half of one percent. It is possible that rare, protein-altering variants also contribute to left-handedness, with larger effects on carriers than the common variants studied so far[32]. One study in an extended, consanguineous family with numerous left-handers did not identify such effects using exome sequencing[33]. In addition, an exploratory study of right-hemisphere language dominance – a trait with roughly 1% frequency that occurs mostly in left-handers – found tentative evidence for rare genetic contributions that implicated the actin cytoskeleton[34], but that study was based on fewer than 100 unrelated participants. Therefore, rare coding variation has yet to be explored in large-scale studies of human handedness. Identifying rare, coding effects on left-handedness may help to elucidate mechanisms of left-right axis development in the human brain.

Meta-analyses have indicated that left-handedness occurs at increased rates in neurodevelopmental disorders, including autism[35] and schizophrenia[36]. In addition, large-scale studies have found that various aspects of structural brain asymmetry are subtly altered in autism[37] and schizophrenia[38]. These associations suggest that population-typical asymmetries are linked to neurotypical function. Both autism and schizophrenia have shown genetic overlaps with structural brain asymmetry, in terms of common variant effects[17,39]. Rare, coding variants are also known to be involved in the genetic architecture of neurodevelopmental disorders[40,41]. While most left-handers do not have these disorders, the increased rates of left-handedness in these disorders suggests a minority of left-handedness might arise from rare, protein-coding variants.

A significant genetic correlation has also been reported between left-handedness and Parkinson's disease, based on common genetic variants[28]. This genetic correlation is at least partly driven by a locus on chromosome 17q21 that spans *MAPT* and other neighboring genes[28]. *MAPT* mutations are a known cause of frontotemporal dementia with parkinsonism, and various other neurodegenerative diseases can involve aberrant aggregation of MAPT within neurons[27].

Here we made use of exome sequence data from the UK Biobank to investigate the contribution of rare, coding variants to left-handedness. We used gene-based analysis to scan for individual genes associated with left-handedness, as well as burden heritability regression[42] to estimate the total, exome-wide contribution of this class of genetic variation. We also queried the extent to which genes that have shown significant associations with schizophrenia, autism, Parkinson's disease or Alzheimer's disease in previous large-scale exome sequencing studies might show associations with left-handedness in the present study.

## Results
### Exome-wide association scan for left-handedness
After sample-level quality control (Methods) there were 313,271 right-handed and 38,043 left-handed individuals (Supplementary Table 1). The rate of left-handedness can vary from roughly 2% to 14% in different regions of the world, which is thought primarily to reflect enforced right-hand use in some cultures[2,3,5,6,26]. To avoid confounding our genetic association analysis, we defined four separate, genetically homogeneous groups of UK Biobank individuals that correspond to major world ancestries, using a combination of self-reported ethnicity and data-driven genetic clustering: Asian ancestry, Black ancestry, Chinese ancestry, and White ancestry (see Methods and Supplementary Fig. 1). As expected, the rate of left-handedness varied between these clusters (Table 1). Within each cluster separately we would then test the association of genetic variants with handedness, and finally meta-analyze across clusters.

There were also 6,511 individuals who reported using both hands equally (Table 1), but this trait was previously found to have poor repeatability in UK Biobank individuals who reported their handedness on more than one occasion[1]. We excluded this group from our genetic association and heritability analyses.

We focused on exonic variants with frequencies ≤1% that met sequence quality criteria (Supplementary Table 2, Supplementary Fig. 2) and had high likelihoods of affecting protein function (Methods; Supplementary Table 3). This "strict" set of variants included frameshift and stop mutations that affect canonical gene transcripts outside of the 5% tail ends of the corresponding proteins, and missense variants when they had Combined Annotation Dependent Depletion (CADD) phred-scaled scores > 20. CADD scores indicate the deleteriousness of genetic variants based on diverse genomic features derived from surrounding sequence context, gene model annotations, evolutionary constraint, epigenetic marks and functional predictions[43]. We also defined a more inclusive, "broad" set of variants that included all "strict" variants plus other variants predicted to have less substantial deleterious effects on protein function (for missense variants this meant CADD phred scores > 1), again with frequencies ≤ 1% (see Methods; Supplementary Table 3).

Separately for the strict and broad variant sets, we ran gene-based association analysis with handedness (left-handed versus right-handed) using an additive burden framework, where each individual's number of minor alleles in a given gene was summed to compute a burden score for that individual and gene[44]. Summary statistics were then meta-analyzed per gene across ancestry groups. There were 18,381 genes analyzed for the strict variant set and 18,925 genes for the broad variant set. Q-Q plots indicated appropriate control of type 1 error (Fig. 1).

For both the strict and broad set of variants, one gene showed statistically significant association with handedness after multiple testing correction (Methods): the gene encoding microtubule component beta-tubulin TUBB4B, with association beta = 1.07, $P = 9.9 \times 10^{-7}$ for the strict set, and beta = 1.06, $P = 1.2 \times 10^{-6}$ for the broad set (Fig. 1, Supplementary Fig. 3). Other genes that showed suggestive evidence for association (with nominal association $P$ values $< 1 \times 10^{-5}$) for the strict and broad sets are shown in Supplementary Table 4. These included the gene that encodes TRAK1, involved in mitochondrial trafficking within axons and associated with neurodevelopmental delay and seizures[45], and myotubularin phosphatase MTMR6, involved in secretion and autophagy[46]. The full exome-wide, gene-based meta-analysis results shown in Fig. 1 are provided in the accompanying Source Data file.

For each of 48 genes implicated in left-handedness by the largest previous genome-wide association study based on common genetic variants[21], we queried our exonic rare-variant association results from the present study. None of these genes showed significant rare-variant associations after Bonferroni correction for 48 tests, for either the strict or broad variant sets (Supplementary Table 5). The most significant individual result among these 48 genes was for *FOXN2* and the strict variant set, with beta=0.33, $P = 0.0067$ (un-corrected). This gene encodes a transcription factor involved in cutaneous and thymic epithelial cell development, but also embryonic central nervous system development (recessive mutation can cause anencephaly and spina bifida)[47].

### Rare *TUBB4B* variants in the general population

For *TUBB4B*, the strict variant set comprised 20 variants in 29 left-handed carriers, and 53 variants in 89 right-handed carriers (Fig. 2). As left-handers comprised 10.8% of the individuals tested, their rate of rare, deleterious *TUBB4B* variants (0.076%) was 2.7 times higher than in right-handers (0.028%). All *TUBB4B* variants were heterozygous, and only one individual carried two different variants, such that many of the variants were present uniquely in single individuals (Fig. 2). Only one *TUBB4B* variant was additionally included in the broad set that was not already included in the strict set (causing amino acid change

## Table 1 | The numbers of individuals in each genetically-informed ancestry cluster, separately by self-reported handedness

| Ancestry cluster | Right-handed | Left-handed | Both hands equally | Total |
|---|---|---|---|---|
| Asian | 6421 (91.1%) | 433 (6.1%) | 198 (2.8%) | 7052 |
| Black | 5237 (91.4%) | 381 (6.7%) | 111 (1.9%) | 5729 |
| Chinese | 1178 (93.3%) | 52 (4.1%) | 33 (2.6%) | 1263 |
| White | 300,435 (87.4%) | 37,177 (10.8%) | 6,169 (1.8%) | 343,781 |

Percentages are given within ancestry clusters

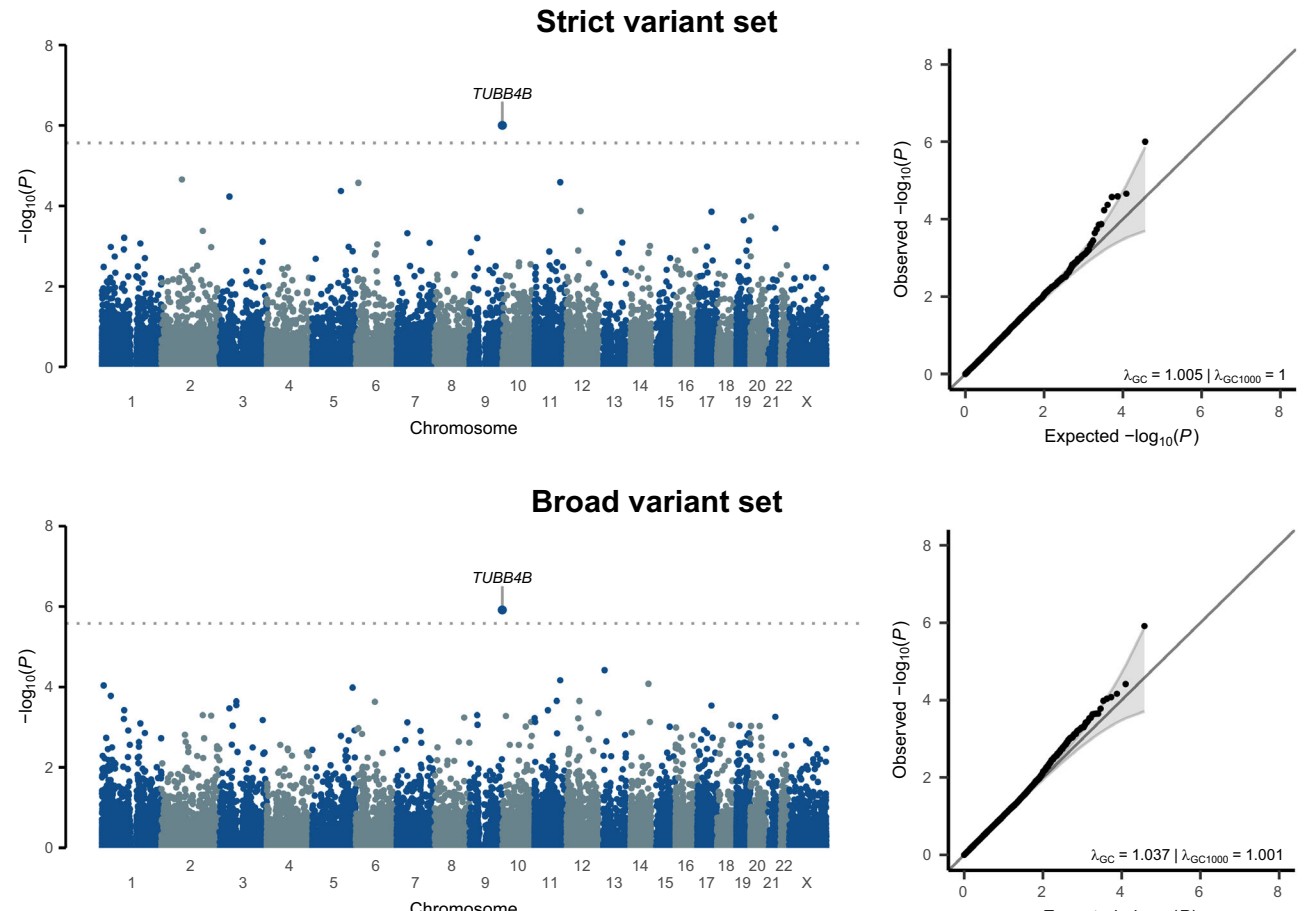

## Strict variant set

## Broad variant set

**Fig. 1 | Exome-wide, gene-based association testing in 38,043 left-handed and 313,271 right-handed individuals, based on rare protein-altering variants.** Top: the strict variant set. Bottom: the broad variant set (that also included the strict variants for this analysis). Left: Manhattan plots show the genome along the x-axis and the gene-wise association significance levels on the y-axis. Dashed lines indicate Bonferroni-based multiple testing correction thresholds. Right: Q-Q plots corresponding to the Manhattan plots (gray shaded areas show 95% confidence intervals for the expected distributions). The exome-wide gene-based association meta-analysis results are provided in a Source Data file.

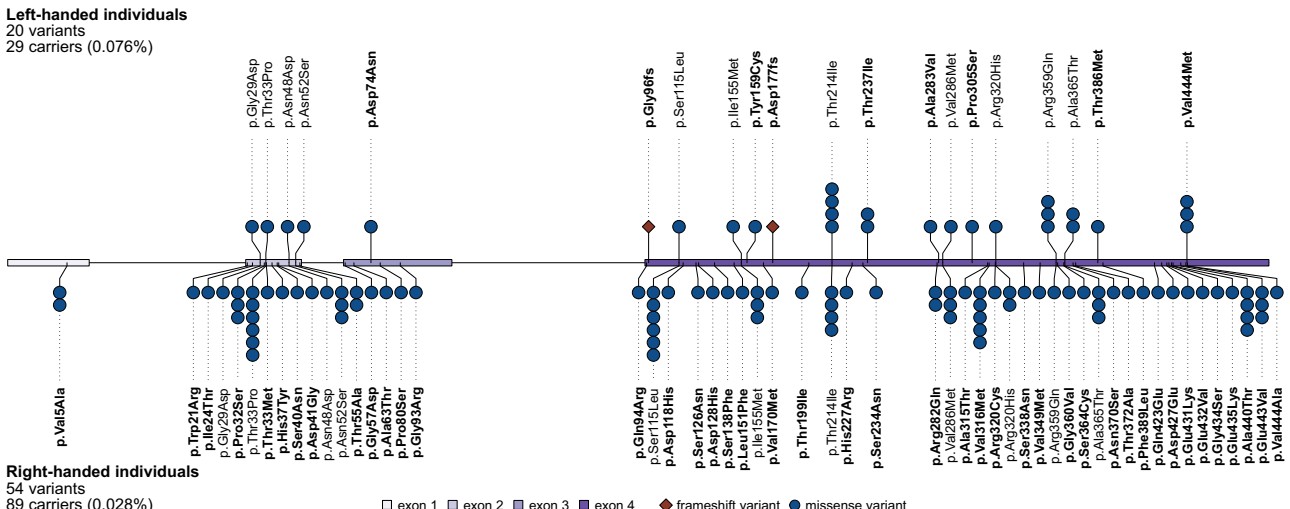

**Fig. 2 | Rare protein-altering *TUBB4B* variants found in left-handed individuals (top) and right-handed individuals (bottom).** Labels in bold indicate variants uniquely observed in either left- or right-handed individuals. The allele count for a given variant is indicated by the number of symbols (circles or diamonds). All variants were heterozygous, and only one individual carried two different variants. Exons and introns of the genomic locus are indicated in the central schema. All but one of the variants (Asp427Glu) met the "strict" criteria for being deleterious. Frameshift mutations were only found in left-handers. Left-handers comprised 10.8% of the individuals in the analyzed dataset, but 32.6% of *TUBB4B* variant carriers.

Asp427Glu), which explains the similarity of strict and broad results for this particular gene. The *TUBB4B* association with left-handedness was therefore driven by variants that are strongly predicted to be disruptive and deleterious. Strict and broad results were sometimes more divergent for other genes (Fig. 1; Supplementary Table 4).

Most of the *TUBB4B* variants caused missense changes, i.e. substituting one amino acid for another at a given point in the protein sequence (Fig. 2, Supplementary Figs. 4 and 5). However, there were two frameshift variants in left-handers, and none in right-handers (Fig. 2), despite right-handers out-numbering left-handers by roughly 8:1. Frameshift variants disrupt the triplet reading frame of DNA and result in mis-translation of protein sequence from that point onwards. As such, they are among the most disruptive types of coding variant. Both of the frameshift variants that we found were predicted to cause degradation of the *TUBB4B* RNA transcript by nonsense-mediated decay (Methods). Therefore, both frameshift variants are likely to lead to haploinsufficiency.

Human TUBB4B shows extremely high conservation of amino acid identities with its orthologs in other vertebrates: 100% homology in chimpanzees, macaques, mice, cattle and dogs, 99.8% in rats, 99.6% in chickens, and 99.3% in clawed frogs (Methods). This high degree of conservation likely contributed to CADD predictions of deleteriousness for many of the *TUBB4B* missense variants, as the gene appears largely intolerant to variation across vertebrates. As regards other, paralogous beta-tubulin genes, humans have eight of these: *TUBB*, *TUBB1*, *TUBB2A*, *TUBB2B*, *TUBB3*, *TUBB4A*, *TUBB6* and *TUBB8*. For each "strict" variant in *TUBB4B* we assessed whether it alters the protein sequence at a site that varies between human beta-tubulin paralogs, or is fully conserved across these paralogs (Supplementary Fig. 4). The two frameshift mutations were counted among those that affect conserved sites. 15 out of 29 left-handed carriers (52%) had variants that changed sites which are conserved across all paralogs, whereas 31 out of 89 right-handed carriers (35%) had such variants (Fig. 2, Supplementary Fig. 4). This suggests that *TUBB4B* variants observed in left-handers tend to affect especially critical sites, although the difference was not statistically significant (chi-square = 2.62, one-tailed $P = 0.052$).

Other rare, heterozygous *TUBB4B* missense variants, and also an in-frame ten amino acid duplication, are known to cause sensorineural and/or ciliopathic disorders which can involve infant blindness and early onset hearing loss[48–50] (see Discussion). However, none of the

specific TUBB4B amino acid changes that we identified in the UK Biobank were reported in previous clinical genetic studies, and neither were any frameshift mutations. We tested whether *TUBB4B* variant carriers in the UK Biobank showed group-average differences from the rest of the dataset in terms of speech reception thresholds or visual acuity, or in the frequency of hearing problems, use of a hearing aid, eye problems or use of glasses, while controlling for age and sex (Methods). No associations were significant (all *P* values > 0.25), which indicates that hearing and vision are not generally affected by the *TUBB4B* coding variants found in the UK Biobank population dataset (Supplementary Table 6).

### Gene-based burden heritability of left-handedness

To quantify the heritability of left-handedness attributable to the genome-wide burden of rare, exonic variants, we applied burden heritability regression[42] (Methods). This analysis was performed only in the genetically-informed "White" ancestry cluster, as this was the only cluster with a sufficiently large number of individuals for this type of analysis (300,435 right-handed and 37,177 left-handed; Table 1). Variants were stratified by strict versus broad annotations (for this specific analysis the strict variants were removed from the broad set in order to distinguish the heritability arising from deleterious versus relatively benign variants). We also stratified into three variant frequency bins, i) minor allele frequency $<1 \times 10^{-5}$; ii) $1 \times 10^{-5} \leq$ minor allele frequency $<1 \times 10^{-3}$); iii) $1 \times 10^{-3} \leq$ minor allele frequency $\leq 1 \times 10^{-2}$.

Aggregated across the three frequency bins and strict and broad variants, the burden heritability best estimate (liability scale) was 0.91% (standard error 0.32%) (Table 2). For "strict" variants the main contribution was from the two rarest frequency bins (minor allele frequencies $<1 \times 10^{-3}$), whereas for "broad" variants those with minor allele frequencies above $1 \times 10^{-5}$ and up to 1% made the main contributions (Table 2). Removing *TUBB4B* from the analysis made little impact on the burden heritability estimates (Table 2), which confirms that the population-level, exome-wide burden heritability involves multiple additional loci beyond this single gene.

### Rare coding variant associations with neurodevelopmental and neurodegenerative disorders in comparison to left-handedness

Recent large-scale association studies based on rare, coding variants identified 24 genes associated with autism[40], 10 genes associated with

**Table 2 | Burden heritability regression for left-handedness shows the proportion of trait disposition due to rare, exonic variants considered over the whole genome**

| Functional group | Minor allele frequency (MAF) | All genes | | | TUBB4B excluded | | |
|---|---|---|---|---|---|---|---|
| | | Heritability | SE | Genes | Heritability | SE | Genes |
| Strict | MAF < 1 × 10$^{-5}$ | 0.14% | 0.13% | 16891 | 0.16% | 0.13% | 16890 |
| Strict | 1 × 10$^{-5}$ ≤ MAF < 1 × 10$^{-3}$ | 0.32% | 0.14% | 16584 | 0.29% | 0.14% | 16583 |
| Strict | 1 × 10$^{-3}$ ≤ MAF ≤ 1 × 10$^{-2}$ | −0.04% | 0.10% | 6476 | −0.04% | 0.10% | 6476 |
| Strict | Aggregate | 0.41% | 0.24% | NA | 0.41% | 0.24% | – |
| Broad minus strict | MAF < 1 × 10$^{-5}$ | 0.08% | 0.08% | 16952 | 0.09% | 0.08% | 16951 |
| Broad minus strict | 1 × 10$^{-5}$ ≤ MAF < 1 × 10$^{-3}$ | 0.18% | 0.10% | 16508 | 0.18% | 0.10% | 16508 |
| Broad minus strict | 1 × 10$^{-3}$ ≤ MAF ≤ 1 × 10$^{-2}$ | 0.24% | 0.12% | 5787 | 0.24% | 0.12% | 5787 |
| Broad minus strict | Aggregate | 0.50% | 0.17% | NA | 0.51% | 0.17% | – |
| All | Aggregate | 0.91% | 0.32% | NA | 0.92% | 0.32% | – |

Liability-scale heritability estimates are presented separately by variant functional groups and frequency bins, as well as aggregated. For this specific analysis (unlike the gene-based exome-wide association scan), the strict variants were removed from the broad set, in order to distinguish the contributions to heritability from disruptive versus more subtle variants. Results are shown before and after excluding *TUBB4B*. SE: standard error of the heritability estimate. "Genes" refers to the number of genes included in the analysis for a given variant frequency bin.

schizophrenia[41], 4 genes associated with Parkinson's disease[51], and 5 genes associated with Alzheimer's disease[52], at exome-wide significance levels. We queried each of these 43 genes in the rare-variant association results of the present study of left-handedness (Supplementary Tables 7–10). The autism-associated gene *DSCAM* (Down syndrome cell adhesion molecule) showed significant association with left-handedness after Bonferroni correction for 43 tests (strict set, beta = 0.17, $P = 3.6 × 10^{-4}$ uncorrected: broad set, beta = 0.15, $P = 5.5 × 10^{-4}$ uncorrected) (Supplementary Table 7). *DSCAM* is involved in central and peripheral nervous system development, including through affecting the interaction of other autism-linked synaptic adhesion molecules[53]. In addition, the autism-associated gene *FOXP1* showed significant association with left-handedness after Bonferroni correction for 43 tests for the broad set only (beta = 0.17, $P = 2.3 × 10^{-4}$ uncorrected) (Supplementary Table 7). *FOXP1* encodes a transcription factor in which disruptive coding variants are known to cause a developmental disorder that includes intellectual disability, autistic features, speech/language deficits, hypotonia and mild dysmorphic features[54].

None of the genes associated with schizophrenia, Parkinson's disease or Alzheimer's disease showed evidence for association with left-handedness after Bonferroni correction (Supplementary Tables 8–10).

## Discussion

By making use of the large UK Biobank general population dataset for exome-wide screening and burden heritability analysis, our study identified a role for rare, coding variants in left-handedness. At the population level, the heritability of left-handedness due to this class of genetic variant was low, at just under 1%. Nonetheless, the carriers of rare coding variants in *TUBB4B*, and potentially also *DSCAM* and *FOXP1*, appear to have substantially higher chances of being left-handed than non-carriers. Implicating these specific genes in left-handedness provides potential insights into mechanisms of left-right axis formation in the brain, as well as genetic susceptibility to brain disorders.

As mentioned in the Introduction, several other tubulin genes have been implicated in both left-handedness and structural brain asymmetry by large-scale studies of common genetic variation, including *TUBB*, the *TUBA1A/TUBA1B/TUBA1C* cluster on chromosome 12, *TUBB3* and *TUBB4A*, as well as microtubule-associated proteins *MAP2*, *MAPT* and *NME7*[17,21,31]. It is therefore especially striking that in a systematic screen of rare coding variation across the entire exome, *TUBB4B* showed the most significant association with left-handedness in the present study. This finding gives further support for the involvement of microtubules in human brain asymmetry. While the common variants implicating tubulin genes are non-coding and likely to

affect gene expression levels, the coding variants in *TUBB4B* indicate that protein sequence changes of this gene can affect handedness and brain asymmetry.

Other heterozygous missense variants in *TUBB4B* have been linked to sensorineural and/or ciliopathic disorders[48–50], but not the specific coding variants that we found in the UK Biobank. Our findings therefore extend the spectrum of phenotypes associated with rare coding *TUBB4B* variants to include the benign trait of left-handedness. It is thought that some of the heterozygous *TUBB4B* missense variants that cause disorders act in a dominant-negative manner, through altering microtubule stability and dynamics which can affect microtubule growth[48–50]. A dominant-negative effect arises when an altered protein adversely affects its normal, unaltered counterpart within the same heterozygous cell. It is possible that some of the heterozygous missense *TUBB4B* variants in the UK Biobank exert dominant-negative or gain-of-function effects that impact microtubule dynamics, but less substantially than the variants that have been linked to clinical disorders. Interactions with microtubule-associated proteins may also be affected. In addition, the two frameshift-causing variants that we found in left-handers in the UK Biobank suggest that haploinsufficiency of *TUBB4B* might affect brain asymmetry, but again without a clinical phenotype.

One of the many cellular functions of microtubules is in motile cilia, which are organelles that project from the cellular surface and can beat/rotate to produce an extracellular fluid flow[55]. Life on earth is based on L-form amino acids rather than the mirror D-form, and this chirality carries through to the macromolecular scale to influence the structure and movement of cilia[55,56], i.e., they tend to beat/rotate in one particular orientation rather than the other. In the early embryos of many mammalian species, this results in unilateral, leftward fluid flow that can trigger asymmetrical gene expression[55]. Lateralized, downstream developmental programs eventually give rise to asymmetries of visceral organ placement and morphology (of the heart, lungs etc.).

Microtubules support the hair-like structure of cilia and contribute to their motility. Therefore, embryonic motile cilia might seem to provide a potentially common mechanism for the developments of brain and visceral asymmetry. However, the typical left-hemisphere dominances for hand preference and language do not usually reverse in people with *situs inversus* and primary ciliary dyskinesia, a rare genetic condition that involves reversal of the visceral organs on the left-right axis, together with impairment of motile ciliary function[57–60]. Furthermore, mutations in tubulin genes are not known as causes of *situs inversus* of the viscera. Together, these observations suggest a developmental disconnect between brain and visceral asymmetries.

As components of the cytoskeleton, microtubules can also contribute to asymmetries at the whole-cell scale, i.e., create uni-

directional biases in the morphology, position, rotation or migration of cells[61], or the intracellular distributions of organelles[62]. In invertebrates and frog embryos, cellular chirality during embryonic development can induce asymmetrical morphology of certain organs, independently of other developing organs or systems[61,63–69]. Therefore, the cytoskeleton may be a source of left-right axis creation, in addition to motile cilia. An organ-intrinsic, microtubule-based, but non-motile-ciliary mechanism of brain left-right axis formation would fit the human genetic findings that have implicated tubulins such as *TUBB4B* in handedness, and also match with the disconnect between brain and visceral asymmetries[17]. An involvement of non-motile cilia is also possible.

Brain magnetic resonance imaging (MRI) data was only available for 13 of the UK Biobank *TUBB4B* variant carriers (left- and right-handers together), which is too small a sample for reliable association mapping with respect to brain structural or functional asymmetries. Neither of the left-handed frameshift variant carriers had MRI data. Studies that identified other *TUBB4B* mutations in sensorineural and/ or ciliopathic disorders also did not report brain scanning[48–50]. However, mutations in different beta-tubulin genes in humans have been identified as causes of extremely rare neurological disorders[70], and some of those clinical genetic studies did include brain imaging data. Intriguingly, mutations in *TUBB2B* can cause asymmetrical polymicrogyria (many and small folds) of the cerebral cortex[71]. Mutations in *TUBB3* can cause asymmetrical cortical dysplasia and unilateral hypohidrosis (reduced sweating on one side of the body, thought to be linked to disrupted function of the cortex, brain stem, and spine)[72,73]. It may therefore be informative to collect brain MRI data from *TUBB4B* variant carriers in future studies.

More generally, the spectrum of disorders attributable to mutations in alpha- and beta-tubulin isotypes includes features consistent with altered neuronal migration and differentiation, as well as axon guidance and maintenance[70]. The present study suggests that brain left-right axis formation may be another aspect, although as microtubules are multi-functional and essential components of cells, they may also affect asymmetrical brain development through various downstream mechanisms. For instance, assembled microtubules have "plus-ends" and "negative-ends" with specialized functions that contribute to developmental processes such as directional migration of progenitor cells, and axonal/dendritic polarity[74]

As regards *DSCAM* and *FOXP1*, although these genes were not significantly associated with left-handedness in the context of exome-wide multiple testing, they showed evidence for association with left-handedness in a targeted look-up of 43 genes that were previously implicated in autism, schizophrenia, Parkinson's disease or Alzheimer's disease at exome-wide significant levels, by large-scale studies[40,41,51,52]. Specifically, rare-coding variants in *DSCAM* and *FOXP1* were implicated in autism by a multi-cohort, exome-wide analysis that included nearly 12,000 affected individuals[40]. The present study suggests that rare, coding variants in these genes are also relevant to left-handedness, which raises the possibility that altered development of the brain's left-right axis is part of the etiology of autism when caused by *DSCAM* or *FOXP1* mutations.

Mutations or copy-number variants affecting *DSCAM* have also been associated with intellectual disability and schizophrenia, and this gene might additionally contribute to the phenotype of Down's syndrome[75,76]. *DSCAM* encodes a cell surface receptor and cell adhesion molecule. In the nervous system DSCAM affects various neurodevelopmental processes including neuronal migration, axon growth and branching, synapse development and synaptic plasticity[53,76]. In mice, DSCAM contributes to the formation of the spinal locomotor circuit, and is also important in voluntary locomotor control through affecting short-term plasticity and synaptic integration within the motor cortex[77]. Such functions may be relevant to the possible association of *DSCAM* with handedness in humans. The transcription factor

encoded by *FOXP1* is involved in a neurodevelopmental disorder that involves intellectual disability with autistic features, together with language impairment[54]. The speech and language phenotype can include dysarthria, motor planning and programming deficits, and linguistic-based phonological errors[78]. The latter aspects may be especially linked to altered brain asymmetry.

In contrast to neurodevelopmental disorders, we saw no evidence that any of nine genes implicated in adult-onset neurodegenerative diseases (Parkinson's disease or Alzheimer's disease) by large-scale exome studies were associated with left-handedness. As mentioned in the Introduction, a significant genetic correlation has been reported between left-handedness and Parkinson's disease based on common genetic variants[28]. This genetic correlation is at least partly contributed by a region of long-range linkage disequilibrium on chromosome 17q21 that spans the gene encoding MAPT and eleven neighboring genes. The region has an unusually complex genomic architecture, which relates to a common inversion polymorphism that spans almost one megabase[79]. Multiple different common genetic variants within this extended genomic locus are associated with left-handedness[20], brain structural asymmetry[17], and many other structural and functional brain traits[80]. However, in the present study of rare, protein-coding variation, *MAPT* showed no association with left-handedness (see the accompanying Source Data file). Four genes encoding tubulin isotypes that have been implicated in neurodegenerative phenotypes by studies in families or singleton patients (*TUBA4A, TUBB2A, TUBB3, TUBB4A*)[81] also showed no nominally significant associations with left-handedness, on the basis of rare, coding variants (Source Data file).

We found that the heritability of left-handedness that was attributable to rare, coding variants, when considered over the whole exome, was just under 1%. This compares to a common single-nucleotide polymorphism (SNP) based heritability for left-handedness of 1–6% in the UK Biobank[20,21]. For a further comparison, schizophrenia and bipolar disorder have shown liability-scale burden heritabilities of 1.7% and 1.8% respectively, when considering predicted loss-of-function variants specifically[42]. Our finding of a significant exome-wide heritability for left-handedness suggests that more genes will be implicated in this trait by rare variant association mapping in even larger, future studies.

Twin studies have not found effects of shared family environment on brain asymmetries[22,23], and left-handedness has shown only subtle associations with environmental, epigenetic and early life factors that have been studied to date[1,82–84]. Most of the variation in brain and behavioral asymmetries may therefore arise stochastically in early development[1,85]. The low burden heritability and SNP-based heritability of left-handedness, together with the strong population-level bias to right-handedness in the population, suggest that developmental mechanisms for brain asymmetry are largely genetically invariant in the population. This may reflect negative selection of variants in genes involved in brain asymmetry[20,86,87]. A microtubule-based mechanism of brain left-right axis formation would be consistent with this, because microtubules are essential for many other, fundamental cellular functions[88,89]. Accordingly, the TUBB4B protein shows over 99% conservation of amino acid sequence across many vertebrate species.

In conclusion, this study revealed a role for rare, protein-altering variants in human handedness, and provided further evidence that microtubules are involved – possibly through affecting molecular, organelle or cellular chirality early in development. This study also shed light on possible commonalities and differences between rare, coding contributions to left-handedness and brain-related disorders.

## Methods

### Ethics
For this study we used data from the UK Biobank[90,91]. The UK Biobank received ethical approval from the National Research Ethics Service

Committee North West-Haydock (reference 11/NW/0382), and all of their procedures were performed in accordance with the World Medical Association guidelines[29]. Written informed consent was provided by all of the enrolled participants.

The present study was conducted as part of UK Biobank registered project 16066, with Clyde Francks as the principal investigator. The study design and conduct complied with all relevant regulations regarding the use of human study participants and was conducted in accordance to the criteria set by the Declaration of Helsinki, with approval from the Ethics Committee Faculty of Social Sciences, Radboud University Nijmegen.

## Data set
Phenotype data were obtained from data release version 10.1 (available on the UK Biobank research analysis platform (https://ukbiobank.dnanexus.com) since 14 April 2022), and the whole exome sequence data were from release version 12.1 (available on the platform since 29 June 2022). We selected "Handedness (chirality/laterality)" (data-field 1707) as our primary phenotype, which was self-reported according to the question "Are you right or left handed?" (presented on a touchscreen). Possible answers were "right-handed," "left-handed," "use both left and right hands equally" and "prefer not to answer." The latter was treated as missing data. Answers were recorded at a maximum of three visits to a UK Biobank assessment center. We used the handedness reported at the first non-missing instance. For individuals who had reported their hand preference at multiple instances, those who were inconsistent in their reported handedness were excluded.

For all individuals with stable handedness data, we selected additional variables to use as covariates: "Sex" (data-field 31), "Year of birth" (data-field 34), "Country of birth" (data-field 1647), "Part of a multiple birth" (data-field 1777), the first 40 principal components derived from common variant genotype data that capture population ancestry (data-field 22009), and the exome sequencing batch (i.e., a binary variable to indicate whether an individual was sequenced as part of the first 50,000 exome release or subsequent releases, due to a difference in the flow cells used). "Country of birth" and "Part of a multiple birth" could be recorded at multiple instances, and again we set these to missing if individuals reported inconsistent answers.

## Defining ancestry clusters
We first grouped 469,804 individuals with exome data into five ancestry groups according to self-reported ethnic identities in UK Biobank data-field 21000:

- Asian or Asian British (includes sub-fields Indian, Pakistani, Bangladeshi and "any other Asian background").
- Black or Black British (includes sub-fields Caribbean, African, and "any other Black background").
- Chinese (includes only Chinese background).
- White (includes British, Irish, and "any other white background").
- Mixed

Answers of "Do not know," "Prefer not to answer," or "Other" were set to missing. Ethnicity was reported at up to four visits. Individuals were only assigned to one of the five ancestry groups if they had non-missing data for at least one instance, and consistently reported their ethnicity with respect to these five groups if reported at multiple instances. For each of the five self-reported ethnic groups separately, we then applied a Bayesian clustering algorithm in the R package "aberrant" version 1.0[92] to genetic ancestry principal components 1–6 (from data-field 22009). This software seeks to define clusters of datapoints and any outliers from them. The "aberrant" package can only cluster along two dimensions, and was therefore run separately three times for each self-reported ethnic grouping: first on principal components 1&2, then 3&4, then 5&6, with inlier threshold lambda = 40. Individuals in the intersect of all three clusters for a given ethnicity were then assigned to one final genetically-informed cluster for each ethnic group.

For the "mixed" ancestry group we obtained a highly dispersed cluster, and therefore these individuals were excluded. See Table 1, Supplementary Fig. 1 and Supplementary Table 1 for further information.

## Sample-level filtering
There were initially 469,316 individuals with whole exome sequence data, and who consistently reported their handedness, country of birth and whether they were part of a multiple birth. We then applied further individual-level quality control. First, individuals with missing data for one or more covariates defined above were excluded. Then we excluded individuals with discordant self-reported and genetically determined sex, as well as those not included in one of the genetically-informed ancestry clusters as described above. For pairs of related individuals inferred as third-degree relatives or closer (kinship coefficient > 0.0442) based on common variant data[90], we excluded one individual from each pair, prioritizing the removal of right-handed individuals and those present in multiple pairs, but otherwise randomly.

In total, 111,491 individuals were removed by all of these steps together, which left 357,825 remaining individuals. Supplementary Table 1 shows that the majority of exclusions occurred for one of two reasons:

1. 39,170 individuals fell outside of all four of the genetically-informed ancestry clusters that were retained: Asian or Asian British, Black or Black British, Chinese, or White. As the rate of left-handedness varied with ancestry (Table 1), then the excluded sample was expected to differ from the included sample in terms of handedness, and other demographic features that correlate with handedness (see details in Supplementary Table 11).
2. 62,882 individuals were excluded due to being related to another individual at third degree level or higher. As mentioned above, when such a pair of relatives comprised one right-handed and one-left-handed individual, the left-handed individual was retained. This was done to maximize the number of left-handers for statistical power in genetic association analysis. Again, this meant that the excluded sample necessarily differed from the included sample in terms of handedness, and other demographic features that correlate with handedness (see details in Supplementary Table 11).

Of the remaining 357,825 individuals after sample-level filtering, 313,271 were right-handed, 38,043 were left-handed, and 6,511 reported using both hands equally. See Table 1 for a breakdown by ancestry clusters. As mentioned earlier, the "both hands equally" phenotype was not considered in our genetic association and heritability analyses due to a relatively low sample size and poor repeatability, but these individuals were included in our exome sequence pre-processing pipeline, described in the following section.

## Whole exome sequence data and filtering
Whole-exome sequencing was performed by the UK Biobank according to protocols described elsewhere[93,94]. Specifically we made use of data from the original quality functionally equivalent (OQFE) protocol[93]. We successively applied genotype- and variant-level filters to the exome data of the 357,825 individuals that remained after sample-level filtering in the pVCF files[94,95]. First, we only kept variants in the exome sequence target regions (as defined in UK Biobank resource 3803), excluding variants in the 100 base pair flanking regions for which reads had not been checked for coverage and quality metrics in the exome processing pipeline. We also removed any monoallelic variants that arose during merging of the individual-level VCFs. Then, we set individual-level genotypes to no-call if the read depth was <7 for

single-nucleotide variant sites or <10 for insertion-deletion sites, and/or if the genotype quality score (GQ) was <20. Variant-level filtering comprised removal of variant sites with an average GQ across genotypes <35, variant missingness rate > 0.10, minor allele count <1, and/or allele balance for variants with exclusively heterozygous genotype carriers <0.15 for single-nucleotide variants and <0.20 for insertion-deletions. Transition-transversion ratios were calculated prior to, and after, variant-level filtering. Filtered pVCF files were converted to PLINK-format binary files (using plink v1.90b6.26), excluding multi-allelic variants, and then merged per chromosome. For chromosome X, pseudo-autosomal (PAR) regions (PAR1: start – basepair 2781479, PAR2: basepair 155701383 – end) were split off from the rest of chromosome X. Any heterozygous genotypes in the non-PAR chromosome X in males were set to missing. See Supplementary Table 2 for the numbers of variants removed at each quality-control filtering step, and Supplementary Fig. 2 for the distribution of numbers of variants per gene after these filters.

## Functional annotation and masks

Functional annotation of variants in pVCF files was performed using snpEff v5.1d (build 2022-04-19)[96]. Variants were assigned to genes based on their physical positions in the genome, and were assigned descriptive annotations using information derived from the Ensembl database (release 105). Additionally, variants were annotated with Combined Annotation Dependent Depletion (CADD) Phred scores from the database for nonsynonymous functional prediction (dbNSFP) (version 4.3a)[97] using the snpEff toolbox snpSift 5.1d (build 2022-04-19).

We then classified variants for downstream analyses based on their functional annotations. We first defined a "strict" set of variants with the highest confidence for altering protein function and being deleterious. Strict variants had a "High" annotation for affecting a canonical gene transcript outside of the 5% tail end of the corresponding protein (variants of this type include highly disruptive mutations such as frameshifts), or else a "Moderate" annotation for affecting a canonical transcript together with a CADD Phred score of at least 20 (variants of this type are typically protein-altering missense variants that are especially likely to be deleterious) (Supplementary Table 3).

We then defined a more inclusive, "broad" set of variants that included all of the strict variants in addition to several other categories with more equivocal evidence for altering protein function: "High" annotated variants that affected alternative gene transcripts outside of 5% tail ends, "Moderate" annotated variants that affected canonical or alternative gene transcripts with CADD Phred scores of at least 1, and "Modifier" variants that affected canonical or alternative gene transcripts with CADD Phred scores of at least 1 (Supplementary Table 3).

## Gene-based association analysis

We applied gene-based association testing using the regenie software v3.2.5[44], which broadly consists of two steps. First, to fit a whole genome regression model to capture phenotypic variance attributable to common genetic effects, we selected a high-quality subset of genetic markers from UK Biobank genotype array data (data category 263). Single-nucleotide polymorphisms with minor allele frequency ≥1%, Hardy-Weinberg Equilibrium test $p$ value > $1 \times 10^{-15}$ (not for non-PAR chr X), and genotype missingness rate ≤1% were selected using plink (v1.90b6.26). We removed variants with high inter-chromosomal linkage disequilibrium according to Mbatchou et al.[44] and further pruned the data to remove intra-chromosomal linkage disequilibrium ($r^2$ threshold of 0.9 with a window size of 1000 single-nucleotide polymorphisms and a step size of 100 single-nucleotide polymorphisms), leaving 502,765 single-nucleotide polymorphisms for whole-genome model fitting and calculation of leave-one-chromosome-out (LOCO) predictions.

LOCO predictions were used as input in step 2, together with filtered exome data, handedness phenotypes and covariates as defined above. We ran gene-based analysis with the "sum" burden function, with alternative allele frequency threshold of ≤ 1%, and run separately for our strict and broad variant annotation masks (where broad included all strict variants too). Firth likelihood ratio testing was applied in regenie to correct gene $p$ values < 0.05 for the unbalanced left:right handed ratio of the study sample. Genes were tested when at least one variant mapped to a given gene. We then meta-analyzed burden association statistics separately for each gene across the four ancestry groups, using inverse-variance weighted meta-analysis in the METAL software (July 2010 version)[98]. Finally, we applied a Bonferroni-adjusted significance threshold of $2.7 \times 10^{-6}$ to account for testing of 18,381 genes with the strict mask, and $2.6 \times 10^{-6}$ for testing 18,925 genes with the broad mask.

## Burden heritability regression for left-handedness

Burden heritability can be estimated by regressing gene burden trait-association statistics on gene burden scores, where the heritability estimate is proportional to the regression slope, while population stratification and any residual relatedness affect the intercept[42]. As left- and right-handedness are categorical traits in the UK Biobank data, for each autosomal variant we first obtained the allele counts[42] with respect to left- versus right-handedness, as produced from per-variant association analysis in regenie (under an additive model)[44]. We then estimated burden heritability using the BHR v0.1.0 package[42], stratified by three allele frequency bins: i) minor allele frequency <$1 \times 10^{-5}$; ii) $1 \times 10^{-5}$ ≤ minor allele frequency <$1 \times 10^{-3}$); iii) $1 \times 10^{-3}$ ≤ minor allele frequency ≤ $1 \times 10^{-2}$, and also stratified by strict versus broad variant types (for this particular analysis the strict variants were removed from the broad set, unlike for the gene-based association scan described above, where the broad set included all strict variants too). We also used BHR to aggregate the burden heritability across frequency bins and across strict and broad variants. The observed-scale burden heritability estimate was converted to a liability-scale estimate, using a sample prevalence of 11% and a population prevalence of 10.4% for left-handedness[2].

## *TUBB4B* analysis

For the *TUBB4B* gene, we mapped all 62 variants with "strict" functional annotations onto the canonical protein sequence (National Centre for Biotechnology Information reference NP_006079), and also with respect to each of the eight other human beta-tubulin paralogous proteins: TUBB (UQL51120), TUBB1 (NP_110400), TUBB2A (NP_001060.1), TUBB2B (NP_821080), TUBB3 (NP_006077), TUBB4A (NP_001276058), TUBB6 (AAI11375), TUBB8 (NP_817124) (Supplementary Fig. 4). Some variants were present in more than one individual (Fig. 2). We counted how many left-handed and right-handed individuals carried variants that altered variable sites as opposed to conserved sites in the various human beta-tubulin paralogs. The two *TUBB4B* frameshift variants were counted among those that affect conserved sites.

For the two frameshift variants, we also used NMDEscPredictor[99] to predict whether they are subject to degradation by nonsense-mediated decay, through introducing premature stop codons. The frameshift variants were: position 360, −1 and position 604, −4 according to RefSeq transcript NM_006088.

For cross-species comparisons, sequence alignment of human TUBB4B protein (NP_006079) was measured against its orthologs in *Pan troglodytes* (NP_006079), *Macaca mulatta* (NP_006079), *Mus musculus* (NP_006079), *Rattus norvegicus* (NP_006079), *Bos taurus* (NP_006079), *Canis lupus familiaris* (NP_006079), *Gallus gallus* (NP_006079) and *Xenopus tropicalis* (NP_006079), using blastp (https://blast.ncbi.nlm.nih.gov/Blast.cgi?PAGE=Proteins).

We visualized the locations of the 60 "strict" missense changes with respect to the three dimensional structure of the human TUBB4B protein, using MutationExplorer[100] (Supplementary Fig. 5). For this, we input Protein Data Bank model "AF-P68371-F1-model_v4" of the human TUBB4B protein, as generated by AlphaFold[101].

We tested whether TUBB4B variant carriers showed group differences compared to the rest of the UK Biobank individuals for several continuous or categorical traits related to vision and hearing: Speech Reception Threshold, i.e., the signal-to-noise ratio at which half of presented speech could be understood correctly (UK Biobank fields 22219 (left ear) and 20021 (right ear)); Visual acuity (fields 5187 (left eye) and 5185 (right eye)); Hearing difficulties/problems (field 2247); Hearing aid user (field 3393); Eye problems/disorders (field 6148); Wears glasses or contact lenses (2207). These tests were performed using general linear modeling for continuous traits and binomial regression for categorical traits, controlling for age and sex in R software v4.2.1 (https://www.R-project.org/) (Supplementary Table 6).

### Statistics and reproducibility

No statistical method was used to predetermine the sample size. Rather, the sample included all available participants from the UK Biobank who fulfilled the various criteria detailed in the Methods sections above (Data set, Defining ancestry clusters, Sample-level filtering). Individuals who did not meet the criteria specified and explained in those three Methods sections were excluded. Randomization and blinding were not performed for this observational study.

### Reporting summary

Further information on research design is available in the Nature Portfolio Reporting Summary linked to this article.

## Data availability

The primary data used in this study are from the UK Biobank[29,90,91,93,94]. The individual-level data can be provided by UK Biobank pending scientific review and a completed material transfer agreement. Requests for the data should be submitted to the UK Biobank: www.ukbiobank.ac.uk. UK Biobank data field/resource/category codes were: Handedness (chirality/laterality) (data-field 1707), "Sex" (data-field 31), "Year of birth" (data-field 34), "Country of birth" (data-field 1647), "Part of a multiple birth" (data-field 1777), genetic ancestry principal components (data-field 22009), self-reported ethnic identities (data-field 21000), "Speech Reception Threshold" (fields 22219 (left ear) and 20021 (right ear)), "Visual acuity" (fields 5187 (left eye) and 5185 (right eye)), "Hearing difficulties/problems" (field 2247), "Hearing aid user" (field 3393), "Eye problems/disorders" (field 6148), "Wears glasses or contact lenses" (field 2207), exome sequence target regions (resource 3803), exome sequence data (category 170), genotype array data (category 263). Other data sources were: Ensembl database (release 105): http://dec2021.archive.ensembl.org/index.html. Database for nonsynonymous functional prediction (dbNSFP) (version 4.3a)[97]. National Centre for Biotechnology Information https://www.ncbi.nlm.nih.gov/ reference sequence for human *TUBB4B* transcript NM_006088, and human proteins TUBB4B (NP_006079), TUBB (UQL51120), TUBB1 (NP_110400), TUBB2A (NP_001060.1), TUBB2B (NP_821080), TUBB3 (NP_006077), TUBB4A (NP_001276058), TUBB6 (AAI11375), TUBB8 (NP_817124). Also TUBB4B in *Pan troglodytes* (NP_006079), *Macaca mulatta* (NP_006079), *Mus musculus* (NP_006079), *Rattus norvegicus* (NP_006079), *Bos taurus* (NP_006079), *Canis lupus familiaris* (NP_006079), *Gallus gallus* (NP_006079) and *Xenopus tropicalis* (NP_006079). The human TUBB4B model was obtained from the Alphafold Protein Structure Database (https://alphafold.ebi.ac.uk/entry/P68371). Our meta-analyzed gene-based association summary statistics (shown in Fig. 1) accompany this paper as a Source Data file. Source data are provided with this paper.

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

## Acknowledgements

The study was conducted using the UK Biobank resource under application no. 16066 with C.F. as the principal applicant. This research was funded by the Max Planck Society (Germany) and the Netherlands Organization for Scientific Research (Language in Interaction consortium: Gravitation grant number 024.001.006 2022.LiL.PD.MPI.023 (to C.F.)). The funders had no role in study design, data collection and analysis, the decision to publish or preparation of the manuscript. Thanks to Carsten Janke for thoughts on the *TUBB4B* mutation analysis.

## Author contributions

Conceptualization: D.S., S.E.F., C.F. Data curation: D.S., S.S.N. Formal analysis: D.S. Funding acquisition: C.F., S.E.F. Investigation: D.S., C.F. Methodology: D.S., S.S.N., C.F. Project administration: C.F. Resources: C.F., S.E.F. Software: D.S. Supervision: C.F. Visualization: D.S., C.F. Writing – original draft: D.S., C.F. Writing – review & editing: D.S., S.S.N., S.E.F., C.F.

## Funding

## Competing interests

The authors declare no competing interests.

## Additional information

**Peer review information** : *Nature Communications* thanks Ammar Al-Chalabi and Sebastian Ocklenburg for their contribution to the peer review of this work. A peer review file is available.

