## [Peer Review File · Nature Communications]

Exome-wide analysis implicates rare protein-altering variants
in human handednessReviewer #1 (Remarks to the Author):

Review of NCOMMS-23-33334-T "Exome-wide analysis implicates rare protein-altering variants in human handedness" by Dick Schijven et al.

1. Questions from the Nature Communications Review System

1.1 What are the noteworthy results?

In their manuscript entitled "Exome-wide analysis implicates rare protein-altering variants in human handedness" Dick Schijven et al. provide the first study on the effect of rare protein-coding variants on human handedness. This paper adds an important puzzle piece to the literature, since the phenotypic variance explained by common genetic variants in GWAS studies is much lower than that explained by genetic effects in behavioral twin studies. The study shows that it is important to also consider rare variants. Moreover, the variants identified provide important insights in the functional pathways involved in handedness ontogenesis. This is a strong article that makes a major contribution to the field and deserves publication in a high impact journal like Nature Communication.

1.2 Will the work be of significance to the field and related fields? How does it compare to the established literature? If the work is not original, please provide relevant references.

Yes, I am sure this work will have a major significance in the field, for the reasons mentioned above.

1.3 Does the work support the conclusions and claims, or is additional evidence needed?

Yes, I think the work supports the conclusions and claims. No additional evidence is needed to make these claims.

1.4 Are there any flaws in the data analysis, interpretation and conclusions? Do these prohibit publication or require revision?

No, I could not identify any flaws that prohibit publication. I only have very few minor comments that the authors will be able to address in a minor revision.

1.5 Is the methodology sound? Does the work meet the expected standards in your field?

Yes, this is a well powered study that is state-of-the-art.

1.6 Is there enough detail provided in the methods for the work to be reproduced?

Yes, the level of detail is sufficient.

2. Check of the reporting summary

The reporting summary seems to be fine to me.

3. Additional comments

Introduction

The authors may wish to consider the following relevant recently published papers in the introduction:

Odintsova VV, van Dongen J, van Beijsterveldt CEM, Ligthart L, Willemsen G, de Geus EJC, Dolan CV, Boomsma DI. Handedness and 23 Early Life Characteristics in 37,495 Dutch Twins. *Twin Res Hum Genet.* 2023 Jun;26(3):199-208.

Odintsova VV, Suderman M, Hagenbeek FA, Caramaschi D, Hottenga JJ, Pool R; BIOS Consortium; Dolan CV, Ligthart L, van Beijsterveldt CEM, Willemsen G, de Geus EJC, Beck JJ, Ehli EA, Cuellar-Partida G, Evans DM, Medland SE, Relton CL, Boomsma DI, van Dongen J. DNA methylation in peripheral tissues and left-handedness. *Sci Rep.* 2022 Apr 4;12(1):5606. doi: 10.1038/s41598-022-08998-0

Song Y, Lee D, Choi JE, Lee JW, Hong KW. Genome-wide association and replication studies for handedness in a Korean community-based cohort. *Brain Behav.* 2023 Sep;13(9):e3121. doi: 10.1002/brb3.3121

Reference list:

Please check the reference list for correctness and adjust if necessary.

For example, I noticed that references 2 and 41 refers to the same paper, 2 correctly, 41 was missing the pages. Also, the reference list is incoherent at times, e.g., most papers have the journal name written out completely, but Reference 7 is abbreviated ("Nat Rev Neurosci.") and many more small issues. In general, the authors should follow the journals reference style in the revised version which seems to be different from what is currently used:

De, S. & Olson, R. Crystal structure of the *Vibrio cholerae* cytolysin heptamer reveals common features among disparate pore-forming toxins. *Proc. Natl Acad. Sci. USA* 108, 7385–7390 (2011).

Suggestion:

Altogether, this a strong manuscript in a well-powered sample with state-of-the-art methodology that adds an important puzzle piece to the literature on the ontogenesis of functional hemispheric asymmetries. I could clearly see this published in *Nature Communications* and could not identify any reasons that would prohibit publication. Therefore, my suggestion is:

"Minor revisions"

Reviewer #2 (Remarks to the Author):

In this interesting paper, Dick Schijven and colleagues test the hypothesis that rare exonic variants influence self-reported handedness in the UK Biobank samples. They report evidence that such variants in the TUBB4B gene are associated with handedness.

Genetic determinants of handedness have been hard to establish because of the difficulty in defining the phenotype, but large datasets are now allowing this question to be answered. It has importance in relation to understanding the basis of neural symmetry, language and neural development. While common variants have been identified previously, this work is novel in that it examines rare variants.

The methods are well described and appropriate. The results correspond to the methods and the findings support the conclusions. I have only a few comments:

Introduction:

Several of the genes involved are implicated in neurodegeneration, and there is an overlap of neurodegenerative disease and schizophrenia genetically. Is there any indication of handedness being implicated in neurodegeneration risk?

Discussion:

Dominant negative and loss of function is discussed as a mechanism, but although a gain of function mechanism is not excluded by the present work, it is not mentioned.

Rare variants in TUBA4A are implicated in multiple populations in amyotrophic lateral sclerosis, an adult-onset neurodegenerative disease. What was the association in this gene by burden testing?

Are any of the variants likely to result in changes to methylation?

Methods:

About a quarter of people were excluded during quality control. What was the bias in this sample? For example, was the age distribution, sex distribution and ancestry similar to that of the remaining population?

Is more information available on the ancestry classification? Did Asian in fact mean South Asian?
How would someone from Japan or Korea be classified?

Responses to reviews, NCOMMS-23-33334-T, Schijven et al. "Exome-wide analysis implicates rare protein-altering variants in human handedness".

Reviewer #1 (Remarks to the Author):

Review of NCOMMS-23-33334-T "Exome-wide analysis implicates rare protein-altering variants in human handedness" by Dick Schijven et al.

1. Questions from the Nature Communications Review System

1.1 What are the noteworthy results?

In their manuscript entitled "Exome-wide analysis implicates rare protein-altering variants in human handedness" Dick Schijven et al. provide the first study on the effect of rare protein-coding variants on human handedness. This paper adds an important puzzle piece to the literature, since the phenotypic variance explained by common genetic variants in GWAS studies is much lower than that explained by genetic effects in behavioral twin studies. The study shows that it is important to also consider rare variants. Moreover, the variants identified provide important insights in the functional pathways involved in handedness ontogenesis. This is a strong article that makes a major contribution to the field and deserves publication in a high impact journal like Nature Communication.

1.2 Will the work be of significance to the field and related fields? How does it compare to the established literature? If the work is not original, please provide relevant references.

Yes, I am sure this work will have a major significance in the field, for the reasons mentioned above.

1.3 Does the work support the conclusions and claims, or is additional evidence needed?

Yes, I think the work supports the conclusions and claims. No additional evidence is needed to make these claims.

1.4 Are there any flaws in the data analysis, interpretation and conclusions? Do these prohibit publication or require revision?

No, I could not identify any flaws that prohibit publication. I only have very few minor comments that the authors will be able to address in a minor revision.

1.5 Is the methodology sound? Does the work meet the expected standards in your field?

Yes, this is a well powered study that is state-of-the-art.

1.6 Is there enough detail provided in the methods for the work to be reproduced?

Yes, the level of detail is sufficient.

2. Check of the reporting summary

The reporting summary seems to be fine to me.

AUTHORS: Thank you - we are very grateful for this positive assessment.

3. Additional comments

Introduction

The authors may wish to consider the following relevant recently published papers in the introduction:

Odintsova VV, van Dongen J, van Beijsterveldt CEM, Ligthart L, Willemsen G, de Geus EJC, Dolan CV, Boomsma DI. Handedness and 23 Early Life Characteristics in 37,495 Dutch Twins. *Twin Res Hum Genet.* 2023 Jun;26(3):199-208.

Odintsova VV, Suderman M, Hagenbeek FA, Caramaschi D, Hottenga JJ, Pool R; BIOS Consortium; Dolan CV, Ligthart L, van Beijsterveldt CEM, Willemsen G, de Geus EJC, Beck JJ, Ehli EA, Cuellar-Partida G, Evans DM, Medland SE, Relton CL, Boomsma DI, van Dongen J. DNA methylation in peripheral tissues and left-handedness. *Sci Rep.* 2022 Apr 4;12(1):5606. doi: 10.1038/s41598-022-08998-0

Song Y, Lee D, Choi JE, Lee JW, Hong KW. Genome-wide association and replication studies for handedness in a Korean community-based cohort. *Brain Behav.* 2023 Sep;13(9):e3121. doi: 10.1002/brb3.3121

AUTHORS: We added a citation of the genome-wide association study of handedness in Korean individuals (Song et al. 2023) in the Introduction (page 1):

‘Genome-wide association studies of human handedness in sample sizes of less than 10,000 individuals did not find significantly associated genetic loci ^{26, 27}, but two larger-scale studies ^{20, 28} have been performed based on the UK Biobank adult population dataset ²⁹, which included over 30,000 left-handed and 300,000 right-handed individuals ... An even larger genome-wide association meta-analysis study of human handedness has also been performed, including the UK Biobank in addition to many other datasets, for a total of 194,198 left-handed and 1,534,836 right-handed individuals ²¹ ...’

We also cited Song et al. (2023) in the Results section (page 3) where we mention differences in handedness rates in different world regions:

‘The rate of left-handedness can vary from roughly 2% to 14% in different regions of the world, which is thought primarily to reflect enforced right-hand use in some cultures ^{2, 3, 5, 6, 26}.’

We decided that the studies of early life characteristics and DNA methylation fitted best in the Discussion section (page 7):

‘Twin studies have not found effects of shared family environment on brain asymmetries ^{22, 23}, and left-handedness has shown only subtle associations with environmental, epigenetic and early life factors that have been studied to date ^{1, 82, 83, 84}. Most of the variation in brain and behavioural asymmetries may therefore arise stochastically in early development ^{1, 85}.’

Reference list:

Please check the reference list for correctness and adjust if necessary.

For example, I noticed that references 2 and 41 refers to the same paper, 2 correctly, 41 was missing the pages. Also, the reference list is incoherent at times, e.g., most papers have the journal name written out completely, but Reference 7 is abbreviated (“*Nat Rev Neurosci.*”) and many more small issues. In general, the authors should follow the journals reference style in the revised version which seems to be different from what is currently used:

De, S. & Olson, R. Crystal structure of the *Vibrio cholerae* cytolysin heptamer reveals common features among disparate pore-forming toxins. *Proc. Natl Acad. Sci. USA* 108, 7385–7390 (2011).

AUTHORS: We have updated the reference list with the correct journal format, and manually corrected any issues that arose previously in Endnote.

Suggestion:

Altogether, this a strong manuscript in a well-powered sample with state-of-the-art methodology that adds an important puzzle piece to the literature on the ontogenesis of functional hemispheric asymmetries. I could clearly see this published in *Nature Communications* and could not identify any reasons that would prohibit publication. Therefore, my suggestion is:

“Minor revisions”

AUTHORS: Many thanks again for these positive comments and helpful suggestions.

Reviewer #2 (Remarks to the Author):

In this interesting paper, Dick Schijven and colleagues test the hypothesis that rare exonic variants influence self-reported handedness in the UK Biobank samples. They report evidence that such variants in the TUBB4B gene are associated with handedness.

Genetic determinants of handedness have been hard to establish because of the difficulty in defining the phenotype, but large datasets are now allowing this question to be answered. It has importance in relation to understanding the basis of neural symmetry, language and neural development. While common variants have been identified previously, this work is novel in that it examines rare variants. The methods are well described and appropriate. The results correspond to the methods and the findings support the conclusions. I have only a few comments:

AUTHORS: We appreciate this positive assessment very much – thank you.

Introduction:

Several of the genes involved are implicated in neurodegeneration, and there is an overlap of neurodegenerative disease and schizophrenia genetically. Is there any indication of handedness being implicated in neurodegeneration risk?

AUTHORS: In our gene-based burden multi-ancestry meta-analysis results for handedness (that accompany our paper as Dataset 1), we checked specific genes that showed significant associations with Parkinson’s disease or Alzheimer’s disease in recent large-scale exome-wide studies:

- Holstege et al. (2022). Exome sequencing identifies rare damaging variants in ATP8B4 and ABCA1 as risk factors for Alzheimer’s disease. *Nature Genetics* 54: 1786–1794.
- Makariou et al. (2023). Large-scale rare variant burden testing in Parkinson’s disease. *Brain*, 146: 4622–4632.

In addition, we checked our results in Dataset 1 for four genes encoding tubulin isoforms that have been implicated in neurodegenerative phenotypes by studies in families or singleton patients (TUBA4A, TUBB2A, TUBB3, TUBB4A), as well as the gene encoding MAPT.

We have integrated these look-ups in the revised version of the paper by adding new supplementary tables (9 and 10), together with explanatory text at various points throughout the paper.

Introduction (page 2):

‘A significant genetic correlation has also been reported between left-handedness and Parkinson’s disease, based on common genetic variants²⁸. This genetic correlation is at least partly driven by a locus on chromosome 17q21 that spans MAPT and other neighbouring genes²⁸. MAPT mutations are a known cause of frontotemporal dementia with parkinsonism, and various other neurodegenerative diseases can involve aberrant aggregation of MAPT within neurons²⁷ ... We also queried the extent to which genes that have shown significant associations with schizophrenia, autism, Parkinson’s disease or Alzheimer’s disease in previous large-scale exome sequencing studies might show associations with left-handedness in the present study.’

Results (page 5):

‘Recent large-scale association studies based on rare, coding variants identified 24 genes associated with autism⁴⁰, 10 genes associated with schizophrenia⁴¹, 4 genes associated with Parkinson’s disease⁵¹, and 5 genes associated with Alzheimer’s disease⁵², at exome-wide significance levels. We queried each of these 43 genes in the rare-variant association results of the present study of left-handedness (Supplementary Tables 7-10) ... None of the genes associated with schizophrenia, Parkinson’s disease or Alzheimer’s disease showed evidence for association with left-handedness after Bonferroni correction (Supplementary Tables 8-10).’

Discussion (page 7):

'In contrast to neurodevelopmental disorders, we saw no evidence that any of nine genes implicated in adult-onset neurodegenerative diseases (Parkinson's disease or Alzheimer's disease) by large-scale exome studies were associated with left-handedness. As mentioned in the Introduction, a significant genetic correlation has been reported between left-handedness and Parkinson's disease based on common genetic variants²⁸. This genetic correlation is at least partly contributed by a region of long-range linkage disequilibrium on chromosome 17q21 that spans the gene encoding MAPT and eleven neighbouring genes. The region has an unusually complex genomic architecture, which relates to a common inversion polymorphism that spans almost one megabase⁷⁹. Multiple different common genetic variants within this extended genomic locus are associated with left-handedness²⁰, brain structural asymmetry¹⁷, and many other structural and functional brain traits⁸⁰. However, in the present study of rare, protein-coding variation, MAPT showed no association with left-handedness (Dataset 1). Four genes encoding tubulin isoforms that have been implicated in neurodegenerative phenotypes by studies in families or singleton patients (TUBA4A, TUBB2A, TUBB3, TUBB4A)⁸¹ also showed no nominally significant associations with left-handedness, on the basis of rare, coding variants (Dataset 1).'

In addition, to explore whether there is a phenotypic association between handedness and neurodegenerative disease in the UK Biobank, we extracted diagnosis information of four major neurodegenerative diseases: Alzheimer's disease, Parkinson's disease, Motor Neuron Disease, and Multiple Sclerosis. The analysis was limited to the left- and right-handed individuals used in our primary analysis in this study (i.e. the individuals in our final exome sequencing dataset who also had available data for handedness and the covariates used in our main analysis).

The diagnosis data fields used were 41270 (Diagnoses - ICD10), 41271 (Diagnoses - ICD9), and 20002 (Non-cancer illness code, self-reported). Specific codes used to define affected individuals are indicated per disease below.

Logistic regression via the glm() function in R was used to test the association of each diagnosis separately with handedness, where the model was defined as follows:

Handedness (left/right) ~ diagnosis (case-control) + sex + year of birth + country of birth

Alzheimer's disease

Affected individuals had the following codes:

ICD10: F00 Dementia in Alzheimer's disease; F000 Dementia in Alzheimer's disease with early onset; F001 Dementia in Alzheimer's disease with late onset; F002 Dementia in Alzheimer's disease, atypical or mixed type; F009 Dementia in Alzheimer's disease, unspecified.

ICD9: 3310 Alzheimer's disease.

Self-report: 1263 dementia/Alzheimer's/cognitive impairment.

Resulting in the following numbers per handedness group:

	Right	Left	% Left
Control	407588	42753	9.5%
Affected	1965	191	8.9%

Logistic regression produced the following result:

	Estimate	SE	Z	P
AD diagnosis	-0.015	0.076	-0.19	8.5E-01*

*No association between AD and handedness.

Parkinson's disease

Affected individuals have the following codes:

ICD10: G20 Parkinson's disease

ICD9: 3320 or 3321 Parkinson's disease

Self-report: 1262 Parkinson's disease

Resulting in the following numbers per handedness group:

	Right	Left	% Left
Control	406498	42642	9.5%
Affected	3055	302	9.0%

Logistic regression produced the following result:

	Estimate	SE	Z	P
PD diagnosis	-0.049	0.061	-0.80	4.2E-01*

*No association between PD and handedness.

Motor Neuron Disease

Affected individuals have the following codes:

ICD10: G122 Motor neuron disease

ICD9: 3352 Motor neurone disease

Self-report: 1259 motor neurone disease

Resulting in the following numbers per handedness group:

	Right	Left	% Left
Control	409041	42893	9.5%
Affected	512	51	9.1%

Logistic regression produced the following result:

	Estimate	SE	Z	P
MND diagnosis	-0.049	0.147	-0.33	7.4E-01*

*No association between MND and handedness.

Multiple Sclerosis

Affected individuals have the following codes:

ICD10: G35 Multiple sclerosis

ICD9: 3409 Multiple sclerosis

Self-report: 1261 multiple sclerosis

Resulting in the following numbers per handedness group:

	Right	Left	% Left
Control	407638	42735	9.5%
Affected	1915	209	9.8%

Logistic regression produced the following result:

	Estimate	SE	Z	P
MS diagnosis	0.058	0.073	0.79	4.3E-01*

*No association between MS and handedness.

In summary, we found no evidence for phenotypic associations of left-handedness with any of these four neurodegenerative diseases in the UK Biobank data. We report these phenotypic association results here in the response document where they will be visible to readers of the paper, but we have not integrated these findings into the manuscript itself, as analysis of phenotypic associations was not part of the study design. Also, a lack of phenotypic association at the population level does not exclude the possibility of some rare genetic variants linking handedness and neurodegenerative disease.

Discussion:

Dominant negative and loss of function is discussed as a mechanism, but although a gain of function mechanism is not excluded by the present work, it is not mentioned.

AUTHORS: We agree and have added gain of function as a possible mechanism in the Discussion (page 6):

'It is possible that some of the heterozygous missense TUBB4B variants in the UK Biobank exert dominant-negative or gain-of-function effects that impact microtubule dynamics, but less substantially than the variants that have been linked to clinical disorders.'

Rare variants in TUBA4A are implicated in multiple populations in amyotrophic lateral sclerosis, an adult-onset neurodegenerative disease. What was the association in this gene by burden testing?

AUTHORS: We looked up TUBA4A (ENSG00000127824) in our gene-based burden multi-ancestry meta-analysis results (that accompany the paper as Dataset 1), but there was no evidence of association with handedness (strict variant set, $\beta=0.063$, $P=0.71$; broad variant set, $\beta=0.0639$, $P=0.70$).

From the revised Discussion (page 7):

'Four genes encoding tubulin isoforms that have been implicated in neurodegenerative phenotypes by studies in families or singleton patients (TUBA4A, TUBB2A, TUBB3, TUBB4A)⁸¹ also showed no nominally significant associations with left-handedness, on the basis of rare, coding variants (Dataset 1).'

Are any of the variants likely to result in changes to methylation?

AUTHORS: We do not know of a database or software to interrogate rare, protein-coding variants for associations with methylation, or predicted effects on methylation (beyond whether they change CpG sites). Methylation has been assessed with respect to common genetic variants in the context of methylation quantitative trait loci (mQTLs), and/or with a focus on non-coding regulatory regions, such as promoter regions. Within coding exons, there is also evidence that methylation can affect alternative splicing. However, we feel that there is not a clear basis to mention this in the manuscript as a relevant mechanism for *TUBB4B*, in the absence of more direct information or measurement. The primary mechanism by which rare, protein-altering variants affect phenotypes is via altered protein sequences.

Also, as our focus was on rare, protein-coding variants, we filtered out synonymous variants, and variants within 5'-untranslated and 3'-untranslated regions. These types of variants might affect DNA methylation. Having removed them during our pipeline means that we would not be well positioned to investigate possible effects of rare, exonic variants on methylation at a gene-based level, without first re-running the variant filtering pipeline on the UK Biobank Research Analysis Platform (which involves considerable cost for a process and dataset of this size).

Methods:

About a quarter of people were excluded during quality control. What was the bias in this sample? For example, was the age distribution, sex distribution and ancestry similar to that of the remaining population?

AUTHORS: We have added information in the Methods section and a new supplementary Table (11) on this issue.

From the Methods (page 9):

'In total, 111,491 individuals were removed by all of these steps together, which left 357,825 remaining individuals. Supplementary Table 1 shows that the majority of exclusions occurred for one of two reasons:

1. 39,170 individuals fell outside of all four of the genetically-informed ancestry clusters: Asian or Asian British, Black or Black British, Chinese, or White. As the rate of left-handedness varied across ancestries (Table 1), then the excluded sample was expected to differ from the

included sample in terms of handedness, and other demographic features that correlate with handedness, and this was indeed the case (see details in Supplementary Table 11).

2. 62,882 individuals were excluded due to being related to another individual at third degree level or higher. As mentioned above, when such a pair or relatives comprised one right-handed and one-left-handed individual, the left-handed individual was retained. This was done to maximise the number of left-handers for statistical power in genetic association analysis. Again, this meant that the excluded sample necessarily differed from the included sample in terms of handedness, and other demographic features that correlate with handedness (see details in Supplementary Table 11).'

From the supplement (supplementary Table 11):

		Excluded	Included	Excluded (%)	Included (%)
Handedness	Right	103689	313271	93.6%	87.5%
	Left	5705	38043	5.2%	10.6%
	Both hands equally	1361	6511	1.2%	1.8%
Sex	Female	62635	191802	56.2%	53.6%
	Male	48856	166023	43.8%	46.4%
Country of birth	England	80295	284848	72.9%	79.6%
	Wales	4322	16437	3.9%	4.6%
	Scotland	7602	30196	6.9%	8.4%
	Northern Ireland	527	2360	0.5%	0.7%
	Republic of Ireland	1006	3621	0.9%	1.0%
	Elsewhere	16464	20363	14.9%	5.7%
Ancestry cluster	White	68197	343781	61.2%	96.1%
	Asian	845	7052	0.8%	2.0%
	Black	579	5729	0.5%	1.6%
	Chinese	96	1263	0.1%	0.4%
	Not clustered / mixed cluster	41774	0	37.5%	0.0%
Part of multiple birth	No	100980	350039	97.7%	97.8%
	Yes	2400	7786	2.3%	2.2%
Exome sequencing batch	First 50k	11362	38310	10.2%	10.7%
	All other	100129	319515	89.8%	89.3%
Year of Birth (Mean, SD)	-	1952 (8)	1951 (8)	-	-

Supplementary Table 11. Information on handedness and other variables that were used as covariates or for stratifying the analysis, in individuals excluded during sample-level filtering, versus those remaining after sample-level filtering.

Is more information available on the ancestry classification? Did Asian in fact mean South Asian? How would someone from Japan or Korea be classified?

AUTHORS: We have added information to the Methods section (page 8):

'We first grouped 469,804 individuals with exome data into five ancestry groups according to self-reported ethnic identities in UK Biobank data-field 21000:

- Asian or Asian British (includes sub-fields Indian, Pakistani, Bangladeshi and 'any other Asian background').

- **Black or Black British (includes sub-fields Caribbean, African, and 'any other Black background').**
- **Chinese (includes only Chinese background).**
- **White (includes sub-fields British, Irish, and 'any other white background').**
- **Mixed'**

Breaking the top-level groups down would result in very low sample sizes for some of the Asian and Black sub-fields. We therefore only used the top-level groupings. Nonetheless, as we used these self-reported ethnicities in combination with clustering based on genotype data, we are confident that we defined relatively homogenous groups in terms of genetic ancestry, for rare-variant genetic association analysis.

Reviewer #1 (Remarks to the Author):

The authors have further improved an already strong work, I do not have further comments and can recommend acceptance.

Reviewer #2 (Remarks to the Author):

I am grateful to the authors for taking the time to so comprehensively respond to my comments. I am happy that they have answered all the points well.